# Single-Walled vs. Multi-Walled Carbon Nanotubes: Influence of Physico-Chemical Properties on Toxicogenomics Responses in Mouse Lungs

**DOI:** 10.3390/nano13061059

**Published:** 2023-03-15

**Authors:** Silvia Aidee Solorio-Rodriguez, Andrew Williams, Sarah Søs Poulsen, Kristina Bram Knudsen, Keld Alstrup Jensen, Per Axel Clausen, Pernille Høgh Danielsen, Håkan Wallin, Ulla Vogel, Sabina Halappanavar

**Affiliations:** 1Environmental Health Science and Research Bureau, Health Canada, Ottawa, ON K1A0K9, Canada; silvia.solorio.rodriguez@gmail.com (S.A.S.-R.); andrew.williams@hc-sc.gc.ca (A.W.); 2National Research Centre for the Working Environment, DK-2100 Copenhagen, Denmark; sspoulsen84@gmail.com (S.S.P.); bramknudsen@gmail.com (K.B.K.); kaj@nfa.dk (K.A.J.); pac@nfa.dk (P.A.C.); ped@nfa.dk (P.H.D.); hakan.wallin@stami.no (H.W.); ubv@nfa.dk (U.V.); 3Department of Public Health, University of Copenhagen, 1353 Copenhagen, Denmark; 4National Institute of Occupational Health, 0304 Oslo, Norway; 5Department of Biology, University of Ottawa, Ottawa, ON K1N 6N5, Canada

**Keywords:** nanomaterials, carbon nanotubes, functionalization, transcriptomics, genotoxicity, inflammation, fibrosis, pulmonary toxicity

## Abstract

Single-walled carbon nanotubes (SWCNTs) and multi-walled carbon nanotubes (MWCNTs) are nanomaterials with one or multiple layers of carbon sheets. While it is suggested that various properties influence their toxicity, the specific mechanisms are not completely known. This study was aimed to determine if single or multi-walled structures and surface functionalization influence pulmonary toxicity and to identify the underlying mechanisms of toxicity. Female C57BL/6J BomTac mice were exposed to a single dose of 6, 18, or 54 μg/mouse of twelve SWCNTs or MWCNTs of different properties. Neutrophil influx and DNA damage were assessed on days 1 and 28 post-exposure. Genome microarrays and various bioinformatics and statistical methods were used to identify the biological processes, pathways and functions altered post-exposure to CNTs. All CNTs were ranked for their potency to induce transcriptional perturbation using benchmark dose modelling. All CNTs induced tissue inflammation. MWCNTs were more genotoxic than SWCNTs. Transcriptomics analysis showed similar responses across CNTs at the pathway level at the high dose, which included the perturbation of inflammatory, cellular stress, metabolism, and DNA damage responses. Of all CNTs, one pristine SWCNT was found to be the most potent and potentially fibrogenic, so it should be prioritized for further toxicity testing.

## 1. Introduction

Carbon nanotubes (CNTs) are tubular nanomaterials formed by one or more cylindrical carbon layers [1]. The main types are classified as single-walled carbon nanotubes (SWCNTs) or multi-walled carbon nanotubes (MWCNTs) [1,2]. Their lengths can reach the micrometer range, but the diameters for SWCNTs and MWCNTs are typically around 1–2 nm and 2–100 nm, respectively [1,2]. Because of their unique size-associated properties such as high Young’s modulus, tensile strength, and thermal conductivity values [3], they are utilized in electronics, batteries, catalysis, gas storage, polymer composites, coatings, adhesives, inks, fluorescent markers, biological labels, and drug delivery carriers [1]. As a result, the production of CNTs has increased over the years, potentially increasing occupational exposure during their synthesis, packaging, and transport processes [4]. Since CNTs are poorly soluble nanomaterials with a high aspect ratio, concerns relating to human health risks are warranted and require careful investigation [5].

If inhaled, CNTs may reach the lung alveolar region, where they can persist for months to years, inducing several toxicological responses such as genotoxicity, inflammation, and fibrosis. A 13-week study using Wistar rats exposed to 0.1, 0.4, 1.5, and 6 mg/m^3^ of MWCNTs estimated clearance halftimes of 151, 350, 318, and 375 days, respectively. The study observed pulmonary toxicity in a dose-dependent manner, with granulomas and alveolitis appearing at 6 mg/m^3^ with increased levels of lactate dehydrogenase (LDH) and fibrosis [6]. In another long-term study, the whole-body inhalation exposure of F344 rats to 0.02, 0.2, and 2 mg/m^3^ of an extensively studied uncoated MWCNT (MWCNT-7) 5 days/week for 104 weeks resulted in an increase in the lung weight and the number of inflammatory cells, total protein, LDH and alkaline phosphatase levels in bronchoalveolar lavage fluid (BALF) in a dose-dependent manner. The induction of carcinomas, adenomas, and pre-neoplastic epithelial lesions was also observed in the 0.2 and 2 mg/m^3^ dose groups [7]. 

The genotoxic potential of CNTs has been demonstrated for Mitsui-7 (another frequently used name for MWCNT-7), which was classified as possibly carcinogenic to humans by the International Agency for Research and Cancer (IARC) [8]. A systematic review conducted by the National Institute for Occupational Safety and Health (NIOSH) showed that CNTs can cause granulomas and pulmonary fibrosis [4]. A previous study conducted by the authors of this paper showed that MWCNTs induced genotoxicity, inflammation, and fibrosis in transgenic Muta^TM^ mice exposed to 36 or 109 µg/mouse of Mitsui-7 or to 26 or 78 µg/mouse of NM-401 on day 90 post-exposure [9].

It has been suggested that toxicological responses, especially lung inflammation, are influenced by the physico-chemical properties of CNTs. For example, C57BL/6 mice exposed via a single intratracheal instillation to 18, 54, and 162 μg/mouse of long or short MWCNTs only resulted in strong inflammatory and fibrotic responses in mouse lungs exposed to long MWCNTs on day 28 post-exposure, suggesting that fiber length may be associated with stronger lung inflammatory and lung fibrotic responses [10]. In Wistar rats, the intratracheal instillation of 0.15 or 1.5 mg/kg of SWCNTs induced persistent pulmonary inflammation on day 90 post-exposure, whereas the highest dose of MWCNTs induced acute inflammatory responses on days 1 and 3 post-exposure, which resolved overtime [11], suggesting the number of walls is important for the tubes’ toxicity. In addition to length and wall number, surface functionalization also plays a role in the toxicity of CNTs. The lung inflammation and genotoxicity of MWCNTs of different diameters, lengths, and types of surface functionalization (pristine, hydroxylated, carboxylated, and aminated MWCNTs) were investigated in animals exposed to 6, 18, or 54 μg MWCNT/mouse and sacrificed on days 1, 28, and 92 post-exposure [12]. The results showed that short and thin MWCNTs were more inflammogenic and that the specific surface area determined with the Brunauer–Emmett–Teller (BET) method was a predictor of pulmonary inflammation and inversely correlated to genotoxicity while surface functionalization was a predictor of resolved inflammation on day 28 post-exposure [12]. Other studies have shown certain functionalization groups to be protective. For example, surface carboxylation was shown to reduce the levels of cytokines, inflammatory cells, levels of albumin and LDH in the lungs of mice exposed to 40 μg/mouse of MWCNTs via pharyngeal aspiration [13]. 

In another in vivo study, the expression profile of a 17-gene pro-fibrotic signature (PFS17) was assessed using transcriptomics data from six in vivo studies of mouse lungs exposed to different MWCNTs at different concentrations and post-exposure time points. The results showed pristine MWCNTs to be more pro-fibrotic than the functionalized MWCNTs after 1, 3, and 28 days of exposure at 18, 54, and 168 µg/mouse [14]. Moreover, the potential pro-inflammatory and pro-fibrogenic effects of different functionalized MWCNTs (carboxylate, polyethylene glycol, amine, sidewall amine and polyetherimide groups) were evaluated in vitro and in vivo [15]. In vitro, in THP-1 (human monocytic cell line) and BEAS-2B (human bronchial epithelial cell line) cells exposed to 60 µg/mL of MWCNTs for 24 h, the production of interleukin-1β, transforming growth factor-β1 and platelet-derived growth factor AA increased after exposure to pristine or aminated MWCNTs, whereas functionalization with carboxylic acid or polyethylene glycol groups decreased the production of those pro-fibrogenic cytokines. These results were also validated in C57BL/6 mice exposed to 2 mg/kg of the same functionalized MWCNTs via oropharyngeal instillation and sampled on day 21 post-exposure. Compared with pristine MWCNTs, MWCNTs functionalized with polyetherimide groups induced collagen deposition, whereas carboxylated MWCNTs decreased the extent of fibrosis [15]. Thus, several properties of CNTs including fiber length, diameter, wall number and surface functionalization have all been shown to play a role in their induced toxicity, depending on the tested endpoints. However, more studies are needed to conclusively understand the role of wall numbers and other properties in the toxicity induced by CNTs. 

The objectives of the present study were to (1) determine the influence of the physicochemical properties, such as single or multi-walled structures, and surface functionalization of CNTs on genotoxic, inflammatory, and pro-fibrotic responses and (2) identify the underlying mechanisms associated with pulmonary responses. In brief, female C57BL/6J BomTac mice were exposed to a single dose of 6, 18, and 54 μg/mouse of 12 individual SWCNTs or MWCNTs with different properties. BALF neutrophil influx and DNA damage were evaluated on days 1 and 28 post-exposure. Lung tissue transcriptomic analysis was conducted on samples collected day 1 post-exposure. Time points were selected based on the previously conducted studies, some of which demonstrated that the inflammatory and DNA damage responses induced by MWCNTs at 24 h persisted for up to 28 days post-exposure in C57BL/6 mice [10,12]. The authors have published several papers consistently using the same rodent strain and species, doses, and post-exposure time points, which allowed for the cross-comparison of results obtained over the years. Changes in gene expression were assessed using whole genome microarrays, Ingenuity Pathway Analysis (IPA), weighted gene co-expression network analysis (WGCNA), gene set enrichment analysis (GSEA), and benchmark dose (BMD) modelling. Perturbed canonical pathways, functions, and pro-fibrotic genes were identified, and the relative potency of CNTs to induce pathway perturbations in lungs was determined.

## 2. Materials and Methods

### 2.1. Physico-Chemical Properties of CNTs

A total of eight SWCNTs and four MWCNTs were included in the study. All CNTs were obtained from Chengdu Organic Chemicals Co., Ltd., Chinese Academy of Sciences (Timesnano, Chengdu, China), except for NM-411, which was provided by Thomas Swan & Co., Ltd. (Consett, UK). The CNTs were pristine or surface-modified by the manufacturer, and they were not further modified before use. The SWCNTs exhibited pristine, hydroxyl (-OH), or carboxylic acid (-COOH) surfaces. For the MWCNTs, four different surface types of functionalization were considered: pristine, aminated (-NH_2_), hydroxylated (-OH) and carboxylated (-COOH) MWCNTs. All 12 types of CNTs were extensively characterized in-house via the computerized image analysis of scanning electron microscopy (SEM) micrographs and X-ray fluorescence (XRF). In addition, combustion elemental analysis was performed on a FLASH 2000 Organic Elemental Analyzer from Thermo Scientific at DB Lab A/S (Odense, Denmark), and BET determination was performed by Quantachrome GmbH & Co. KG (Odelzhausen, Germany) (information is provided in Table 1). The lengths of the SWCNTs could not be determined because of their tangled morphology, and the lengths of the MWCNTs ranged from 213.6 to 730.85 nm. The mean diameter of the MWCNTs, as determined from experimental data, spanned from 7.46 to 16.42 nm. The SWCNTs appeared to bundle so strongly that the individual diameters could not be determined. The diameters of the SWCNT bundles were 6.72–24.03 nm, which was set to 1–2 nm. The content of iron oxide (Fe_2_O_3_) was highest in NRCWE-055 (4.39 wt%) and the lowest in NRCWE-062 (0.58 wt%). Relatively high cobalt oxide (CoO) contents were found in NRCWE-062, NRCWE-063 and NRCWE-064 (4.6, 5.87 and 5.51 wt%, respectively). The nickel oxide (NiO) content ranged from 0.04 wt% in NRCWE-055 to 1.97 wt% in NRCWE-061. Magnesium oxide (MgO) was present from 0.01 wt% (NRCWE-061) to 0.177 wt% (NRCWE-057), and manganese oxide (MnO) was present from 0.005 wt% (NRCWE-061) to 0.044 wt% (NRCWE-057). Appendix A shows the physico-chemical properties of the CNTs provided by the manufacturer. 

### 2.2. Material Dispersion

All CNTs were dispersed according to the work of Poulsen et al. [12] and Hadrup et al. [17]. In brief, 2.56 mg/mL stock suspensions were prepared in 0.45 μm filtered Milli-Q Nanopure water (Merck Life Science A/S, Soborg, Denmark) with 2% (*w*/*v*) mouse serum. Stock suspensions were sonicated for 16 min on ice water using a 400 W Branson Sonifier S-450D (Branson Ultrasonics Corp., Danbury, CT, USA) with a disruptor horn (Model number: 101-147-037) and 10% amplitude (10 s on and 10 s off). Thus, the CNTs were sonicated at 40 W for 16 min. Exposure dilutions were prepared immediately after sonication and were sonicated for 2 more min before instillation.

### 2.3. Animal Exposure and Sample Collection

Female C57BL/6J BomTac mice aged 6–7 weeks (19 ± 1.5 g) were acclimatized for at least 7 days and housed in polypropylene cages (3 or 7 animals per cage) with bedding (sawdust) at 21 ± 1 °C, 50 ± 10% humidity and a 12 h light/dark period. Mice had access to food and water ad libitum. All experimental procedures followed the guidelines for the care and handling of laboratory animals established by Danish laws and regulations. The study complied with the EC Directive 86/609/EEC on the use of animals for laboratory experiments and was approved by the Danish ‘Animal Experiments Inspectorate’ (permission 2012-15-2934-00223).

Mice were exposed to CNTs via the single intratracheal instillation of 6, 18, or 54 μg/mouse, with *n* = 36 control animals (12 individual control groups were pooled in the analysis) and *n* = 7 per exposed group. The control group was instilled with Milli-Q Nanopure water with 2% sonicated serum, as described for CNT dispersion [17]. On days 1 and 28 post-exposure, mice were sacrificed and both right and left lung lobes were isolated, cut in pieces and stored at −80 °C. BALF cells were also collected on days 1 and 28 post-exposure to evaluate BALF neutrophil influx and DNA damage. The average weight of the mice on the day of exposure was 19.4 ± 1.2 g.

### 2.4. Comet Assay

The comet assay was conducted according to the work of Poulsen et al. [12] and Jackson et al. [18]. Samples were prepared from 40 μL of re-suspended BALF cells and 60 μL of HAM’s freezing F-12 medium (Thermo Fisher Scientific Inc., Roskilde, Denmark) with 1% penicillin/streptomycin (Thermo Fisher Scientific., Roskilde, Denmark), 15% fetal bovine serum (Thermo Fisher Scientific Inc., Roskilde, Denmark) and 10% dimethyl sulfoxide (Merck Life Science A/S, Soborg, Denmark). BALF cells were thawed at 37 °C, and frozen lung tissue was homogenized in Merchant’s medium. Cells were suspended in agarose and then embedded on Trevigen CometSlides^TM^ (Trevigen, Gaithersburg, MD, USA). Slides were cooled and placed in a lysis buffer overnight at 4 °C. Then, slides were incubated in an alkaline solution for 40 min, and electrophoresis was run at 38 V and 300 mA for 25 min. Slides were neutralized, fixed in ethanol, and placed on a 45 °C plate for 15 min. Cells were incubated with Tris–EDTA-buffered SYBR^®^ Green (Thermo Fisher Scientific Inc., Roskilde, Denmark) fluorescent stain for 15 min and dried at 37 °C. DNA damage was analyzed using the Pathfinder^TM^ system (Imstar, Paris, France) as the mean % DNA in the tail for each animal. Statistically significant differences between exposed animals and controls were determined with a one-way analysis of variance (ANOVA) (Dunnett’s post-hoc) or the Kruskal–Wallis test (Dunn’s post-hoc) using GraphPad Prism 9.2.0 (GraphPad Software, San Diego, CA, USA).

### 2.5. BALF Neutrophil Influx

BALF cellularity was analyzed according to the work of Poulsen et al. [12]. BALF cells were separated from BALF via centrifugation at 400 g for 10 min at 4 °C; then, they were re-suspended in 100 μL of HAM’s F-12 medium (Thermo Fisher Scientific Inc., Roskilde, Denmark) with 1% penicillin/streptomycin (Thermo Fisher Scientific Inc., Roskilde, Denmark) and 10% fetal bovine serum (Thermo Fisher Scientific Inc., Roskilde, Denmark). Cell viability was determined with a NucleoCounter NC-200 TM system (ChemoMetec, Lillerød, Denmark). To determine the number of neutrophils, 40 μL of a cell suspension was centrifuged on microscope slides at 55 g for 4 min and fixed for 5 min in 96% ethanol. The slides were stained with May–Grunwald–Giemsa (Merck Life Science A/S, Soborg, Denmark), and 200 cells per sample were counted with 100× magnification. Statistically significant differences between exposed animals and controls were determined using the Kruskal–Wallis test with Dunn’s post-hoc using GraphPad Prism 9.2.0.

### 2.6. Total RNA Extraction

RNA was extracted from random sections of the frozen lung tissue (left lung) using a Trizol^TM^ reagent (Thermo Fisher Scientific Inc., Carlsbad, CA, USA) and stainless-steel beads for homogenization with a Mixer Mill NM400 (25.0 Hz for 2 min) (Retsch^®^, Haan, Germany). RNA was purified using Direct-zol RNA Miniprep Kits (Zymo Research Corp., Irvine, CA, USA) according to the manufacturer’s instructions. For transcriptomic analysis, *n* = 8 per control group and *n* = 5 per experimental group were randomly selected from the total control and exposed groups. The RNA concentration and purity were determined through 260/280 and 260/230 absorbances, respectively, using a NanoDrop Spectrophotometer 2000 (Thermo Fisher Scientific Inc., Waltham, MA, USA). RNA quality and integrity were determined using the Agilent TapeStation system (Agilent Technologies Inc., Santa Clara, CA, USA). RNA samples showed 260/280 and 260/230 ratios of ~2.0, and RNA integrity numbers were above 5. Samples were stored at −80 °C until they were used for microarray experiments.

### 2.7. Microarrays

The Low Input Quick Amp Labeling Kit (Agilent Technologies Inc., Santa Clara, CA, USA) was used to generate double-stranded cDNA and then fluorescent cRNA according to the manufacturer’s instructions. The method uses a T7 RNA polymerase blend that simultaneously amplifies the target material and incorporates cyanine 3-cytidine triphosphate (Cy3 for RNA reference) or cyanine 5-cytidine triphosphate (Cy5 for RNA samples). The amount of RNA sample and universal mouse RNA reference (Agilent Technologies Inc., Santa Clara, CA, USA) used for the reaction was 200 ng. After synthesis, cRNA was purified using RNeasy Mini Kits according to the protocol provided by the manufacturer (Qiagen N.V., Venlo, the Netherlands). Each cRNA sample and cRNA reference was mixed at an equimolar amount of 300 ng and hybridized to 8 × 60 K Sure Print G3 Mouse gene expression microarrays (Agilent Technologies Inc., Santa Clara, CA, USA) for 17 h at 65 °C with a rotation speed of 10 rpm. Slides were washed and scanned on an Agilent Scanner G2505B (Agilent Technologies Inc., Santa Clara, CA, USA). Expression data were extracted using Agilent Feature extraction software version 11.0.1.1 (Agilent Technologies Inc., Santa Clara, CA, USA).

### 2.8. Statistical Analysis of Microarray Data

The statistical analysis of microarray datasets was conducted as described by Rahman et al. [9]. A reference randomized block design with the samples labeled with Cy5 and the reference labeled with Cy3 was used [19]. The locally weighted scatterplot smoothing (LOWESS) regression modelling method was used to normalize data, and a MicroArray Analysis of Variance (MAANOVA) was conducted in R to determine statistically significant differential gene expression [20]. The Fs-statistic was used to test for treatment effects, and permutation analysis using residual shuffling (nperm = 30,000) was used to estimate *p*-values [21]. The resulting *p*-values were adjusted for multiple comparisons using the false discovery rate and multiple testing corrections [22]. Fold-change calculations were based on the least-square means [23]. The data discussed in this publication were deposited in National Center for Biotechnology Information’s Gene Expression Omnibus (Solorio-Rodriguez et al., 2023) and are accessible through GEO Series accession number GSE223520 (https://www.ncbi.nlm.nih.gov/geo/query/acc.cgi?acc=GSE223520).

### 2.9. Gene Ontology (GO) and Pathway Analysis of Differentially Expressed Genes (DEGs)

DEGs were defined as genes showing a fold-change expression of at least ±1.5 in either direction compared with the time-matched untreated controls and an FDR *p*-value ≤ 0.05. Lists of DEGs were generated with IPA software, and the overlapping DEGs depicted in Venn diagrams were generated using Venn (VIB, UGent Bioinformatics & Evolutionary Genomics, Belgic, http://bioinformatics.psb.ugent.be/webtools/Venn/ (accessed on 1 September 2021)).

The functional enrichment analysis of the DEGs was performed using the Database for Annotation, Visualization, and Integrated Discovery (DAVID) and Bioinformatics (https://david.ncifcrf.gov/ (accessed on 30 September 2021), USA). Based on DEGs with a |fold change| ≥1.5, GO biological processes were considered significant if they showed an FDR *p*-value ≤ 0.05. Canonical pathways were identified using IPA software and were considered significantly perturbed if −log (*p*-value) was ≥1.3 for more than 5 DEGs. Additionally, pathways were considered significantly activated with a z-score greater than 2, whereas pathways with a z-score less than −2 were considered significantly inhibited. The bar graphs and heatmaps were created using GraphPad Prism 9.2.0.

### 2.10. WGCNA and GSEA

WGCNA can be used to find clusters of highly correlated genes, which allows for the identification of key modules [24]. In this study, WGCNA was conducted to identify gene modules whose expression pattern was found in the in vivo pulmonary toxicity induced by different substances such as polycyclic aromatic hydrocarbons, cigarette smoke, diesel exhaust, ozone, particulate matter, bleomycin, radiation, and nanomaterials, among others (Appendix A) [10,25,26,27,28,29,30,31,32,33,34,35,36,37,38,39,40,41,42,43,44,45,46,47,48,49,50,51,52,53,54,55,56,57,58,59,60,61,62,63,64,65]. Once modules were identified from those publicly available transcriptomic datasets, the biological functions of each module were determined, and GSEA was conducted to identify which modules were perturbed after exposure to the 12 CNTs included in this study. The advantage of this approach was to understand and compare the biological response of the 12 CNTs based on specific pulmonary toxicity-related modules.

#### 2.10.1. WGCNA

WGCNA (version 1.70-3) was conducted according to the work of Sutherland et al. [66] in R software. All datasets were obtained from GEO and ArrayExpress (https://www.ebi.ac.uk/arrayexpress (accessed on 14 April 2021)) (Appendix A). Studies using microarray and bead arrays were normalized and processed as described in [67,68]. Fold changes from studies using Illumina HiSeq were estimated using normalization by counts per million. This analysis comprised 43 studies, resulting in a total of 453 experimental conditions in mouse lungs. A default soft threshold of 12 was selected. Modules were unsigned (contained both induced and repressed genes). Average linkage hierarchical clustering was used to build the gene dendrogram, and the modules were created with the dynamic branch cut algorithm (cutreeDynamic). Twenty-three modules containing 10,486 genes were identified.

The enrichment analysis for the biological function of each module was performed through DAVID bioinformatics and IPA software. From DAVID bioinformatics, biological processes, molecular functions, and KEGG (Kyoto Encyclopedia of Genes and Genomes) pathways with more than 5 DEGs and an FDR *p*-value ≤ 0.05 were included in the analysis. From IPA software, canonical pathways with more than 5 DEGs and a −log (*p*-value) ≥ 1.3 were included. Moreover, upstream regulators (Filter: Genes, RNAs, and proteins, with a *p*-value of overlap ≤ 0.05) of DEGs, associated diseases/disorders, and molecular/cellular functions were also identified.

#### 2.10.2. GSEA

GSEA was conducted using the 23 modules identified in the WGCNA on the transcriptomic data generated from the 12 CNT exposures. GSEA was used to rank all genes in a dataset and to calculate an enrichment score (ES) for each gene set, reflecting how often members of that gene set occurred at the top or bottom of the ranked dataset. Then, by normalizing the ES (Normalized Enrichment Score (NES)), GSEA accounted for differences in gene set size. The fgsea R package [69] was used to calculate the NES. Positive NES values were interpreted as gene set enrichment at the top of the ranked list. In contrast, negative NES values were interpreted as gene set enrichment at the bottom of the ranked list. Heatmaps were created using GraphPad Prism 9.2.0.

### 2.11. Pro-Fibrotic Transcriptomic Signature

PFS17 was previously identified and validated by Rahman et al. [14]. Principal component analysis (PCA) was performed using the prcomp function in R using the dataset of publicly available in vivo lung transcriptomic studies described by Rahman et al. [14] (Appendix A) [26,27,42,70,71,72,73,74,75,76,77,78] and the dataset produced in this study. In order to classify the CNTs investigated in this study as pro-fibrotic and non-fibrotic, a Gaussian linear analogy was used to estimate the probability of class membership, with class probabilities greater than 0.8 being considered pro-fibrotic and class probabilities less than 0.2 being considered non-profibrotic. Probabilities between 0.8 and 0.2 classified samples as unknown.

### 2.12. Potency Ranking of CNTs

Transcriptomic datasets were imported into BMDExpress version 2.30. Genes were prefiltered using Williams trend (*p*-value cutoff of 0.5 and fold-change value of 1.5) and were modeled to identify the best dose–response relationships using the following functions: Exponential 2, Exponential 3, Exponential 4, Exponential 5, Linear, Polynomial 2, and the restricted Power model. The BMR factor was set at 1.349 (10%). The post-modelling parameters were Best BMD ≤ highest dose, Best fit (*p*-value ≥ 0.1), Best BMDU/BMDL ≤ 40, Best BMDU/BMD ≤ 20, and BEST BMD/BMDL ≤ 20. Response genes were mapped to Reactome, KEGG, and GO using the Defined Category analysis feature.

Multiple transcriptional potency estimates were examined including the National Toxicology Program (NTP) approach [79] with Reactome pathways, the 25th ranked gene [80], and the lowest consistent response dose (LCRD) [81]. The NTP approach required each pathway to have at least 3 genes with a BMD and that 5% of the pathway consisted of genes with BMDs. The pathway with the lowest median BMD was used as the potency estimate. The 25th ranked gene approach consisted of rank ordering the BMDs from lowest to highest and selecting the 25th ranked gene. The LCRD method was used to identify the most sensitive nonoutlier gene BMD where a consistent response was achieved. The LCRD was determined by ranking the BMDs from lowest to highest and calculating the ratio (rank n + 1/rank n) representing the relative change in the BMDs between each ranked gene. The LCRD was the lowest BMD, where all subsequent relative changes less than 1.66 or the lowest BMD in the “consistent response group of BMDs” as all the sequential BMDs in this group had at least one BMC that was within a ¼ log difference in value.

## 3. Results

### 3.1. DNA Strand Breaks by Comet Assay

DNA strand breaks in BALF cells were evaluated on days 1 and 28 post-exposure. SWCNTs did not induce significant DNA damage at any doses or time points (~4%) compared with the percentage of DNA in the tail observed in the control group (4.19%). In contrast, on day 1, significantly more DNA strand breaks were observed for three individual MWCNTs compared with the controls. NRCWE-062 (MWCNTs-pristine) induced 5.8 and 6.08%, NRCWE-063 (MWCNTs-OH) induced 10.18 and 11.27%, and NRCWE-064 (MWCNTs-COOH) induced 8.8 and 8.9% increases in DNA in the tail after exposure to 6 and 18 μg/mouse, respectively. However, no statistically significant increase was observed in the 54 μg/mouse group (~6% of DNA in the tail) (Figure 1A). On day 28, the control and SWCNT groups showed 3.15% of DNA in the tail. A statistically significant increase in DNA damage was only observed for NRCWE-062 (MWCNTs-pristine), NRCWE-063 (MWCNTs-OH), and NRCWE-064 (MWCNTs-COOH) at 6, 18 or 54 µg/mouse (~5–9% of DNA in the tail), after 28 days of exposure (Figure 1B).

Appendix A shows the percentage of DNA in the tail in lungs on days 1 and 28 post-exposure. The percentage of DNA in the tail in lungs on day 1 post-exposure was 3.3% for the controls and mice exposed to most SWCNTs and MWCNTs, except for NRCWE-054 (SWCNTs-COOH), which showed a slight increase of 4.9% at 6 μg/mouse (Appendix A). On day 28 post-exposure, the percentage of DNA in the tail in lungs for the controls and most CNTs was 5.15%, with only NRCWE-053 (SWCNTs-OH) and NRCWE-054 (SWCNTs-COOH) showing a significant (~12%) increase in DNA in the tail at 6 and 18 μg/mouse, respectively (Appendix A). The % DNA in the tail was statistically significantly decreased for a few samples (NRCWE-055, NRCWE-056, NRCWE-057, and NRCWE-061), which showed values from 1.4 to 3%; however, they were not biologically relevant (Appendix A). 

### 3.2. BALF Neutrophil Influx

The total number of neutrophils in BALF was evaluated on days 1 and 28 post-exposure. On day 1 post-exposure, NRCWE-051 (SWCNTs-pristine) induced a significant increase in neutrophils in the BALF of mice exposed to 6, 18 and 54 μg/mouse compared with other SWCNTs, although, all CNTs increased neutrophil recruitment in a dose-dependent manner. Among the SWCNTs, NRCWE-057, NRCWE-052, and NRCWE-54 showed the highest values (~150,000 cells) at 54 μg/mouse, whereas NM-411, NRCWE-055, and NRCWE-056 showed the lowest response (~50,000 cells) at the same dose. Among the MWCNTs, NRCWE-061 (MWCNTs-NH_2_) and NRCWE-062 (MWCNTs-pristine) induced the highest values of the total number of neutrophils at 18 and 54 μg/mouse. (Figure 2A).

On day 28 post-exposure, the number of neutrophils decreased compared with the levels observed on day 1 post-exposure (from ~150,000 to ~35,000 cells), but the number of neutrophils for the majority still remained significantly higher compared with the controls. Although reduced, on day 28 post-exposure, both NRCWE-051 (SWCNTs-pristine) and NRCWE-061 (MWCNTs-NH_2_) showed the highest response compared with the controls at 18 and 54 μg/mouse, both with similar values from ~12,000 to ~32,000 cells (Figure 2B).

Appendix A show the linear relationship between the total number of neutrophils in BALF and the total deposited surface area of CNTs (BETxdose). The total number of neutrophils on day 1 significantly correlated with the total instilled surface area (Appendix A; *p* < 0.0001), but no correlation was found for each dose group. After 28 days of exposure, the neutrophil counts only significantly correlated at 6 μg/mouse (Appendix A; *p* = 0.0174).

### 3.3. Gene Expression Analysis

A list of DEGs in lung tissue after 1 day of exposure to three different doses (6, 18, or 54 μg/mouse) of individual SWCNTs or MWCNTs, relative to matched vehicle controls, is presented in Appendix A. Figure 3 summarizes the number of up- and downregulated genes for all doses in each CNT group. In general, for all CNTs, the numbers of DEGs in the 6 and 18 μg/mouse dose groups were low. 

Among the SWCNTs, NM-411 (SWCNTs-pristine) only regulated 3, 7, and 26 DEGs at 6, 18, and 54 μg/mouse, respectively. The number of DEGs increased in a dose-dependent manner for NRCWE-051 (SWCNTs-pristine), with 392, 537, and 770 DEGs in the 6, 18, and 54 μg/mouse dose groups, respectively. The other SWCNTs induced the differential expression of a small number of genes at the low doses of 6 and 18 μg/mouse; however, the expression of a large number of genes was observed in the 54 μg/mouse dose group, with 26 to 366 genes showing altered expression levels (Figure 3). NRCWE-51 (SWCNTs-pristine) was the only CNT type that shared a high number of DEGs across the dose groups (Appendix A). A general overview of the individual DEGs in each CNT group is shown in Appendix A. Comparatively, NRCWE-061 (MWCNTs-NH_2_) showed a dose-dependent response, with 22, 93, and 547 DEGs at the 6, 18, and 54 μg/mouse doses, respectively. A total of 92, 51, and 22 DEGs were observed after exposure to 18 μg/mouse of NRCWE-062 (MWCNTs-pristine), NRCWE-063 (MWCNTs-OH), and NRCWE-064 (MWCNTs-COOH), respectively. At 54 μg/mouse, NRCWE-062 (MWCNTs-pristine) regulated 395 DEGs, NRCWE-063 (MWCNTs-OH) regulated 231 DEGs, and NRCWE-064 (MWCNTs-COOH) regulated 217 DEGs (Figure 3). The SWCNTs and MWCNTs induced similar numbers of DEGs at 54 μg/mouse, except for NM-411 (SWCNTs-pristine) with 26 DEGs and NRCWE-051 (SWCNTs-pristine) with 770 DEGs in the high-dose group (Figure 3). 

### 3.4. Canonical Pathways Analysis

The biological processes regulated by CNTs are represented in Appendix A and described in the Appendix A. The canonical pathways considered for the analysis are shown in Appendix A. Figure 4 summarizes the number of significantly perturbed canonical pathways associated with DEGs in each CNT group. Not much transcriptomic activity in terms of perturbed pathways was observed in the low-dose group of 6 μg/mouse except for NRCWE-051 (SWCNTs-pristine), which perturbed canonical pathways in a dose-dependent manner. NRCWE-054 (SWCNTs-COOH), NRCWE-061 (MWCNTs-NH_2_), and NRCWE-062 (MWCNTs-pristine) also induced pathway perturbations at the medium dose of 18 µg/mouse. All CNTs, except for NM-411, perturbed a significant number of pathways at 54 µg/mouse. No perturbed canonical pathways were associated with NM-411-regulated DEGs.

Among the top 15 most significant canonical pathways perturbed after exposure to NRCWE-051 (SWCNTs-pristine) at 6 μg/mouse were agranulocyte/granulocyte adhesion and diapedesis, communication between innate and adaptive immune cells, the role of hypercytokinemia/hyperchemokinemia in the pathogenesis of influenza, IL-10 signaling, and LXR/RXR activation (Figure 5A). In addition, the role of hypercytokinemia/hyperchemokinemia in the pathogenesis of influenza, IL-17 signaling, acute-phase response signaling, the LPS/IL-1-mediated inhibition of RXR function, the unfolded protein response, and tumor microenvironment pathways were also found to be activated, and the erythropoietin signaling pathway was found to be inhibited (Figure 5B). 

At the 18 µg/mouse dose, two SWCNTs (NRCWE-051 and NRCWE-054) and two MWCNTs (NRCWE-061 and NRCWE-062) perturbed common pathways associated with inflammation, such as acute-phase response signaling, agranulocyte/granulocyte adhesion and diapedesis, and IL-17 signaling. However, exposure to NRCWE-051 (SWCNTs-pristine) or NRCWE-054 (SWCNTs-COOH) perturbed a higher number of pathways associated with cellular stress, and xenobiotic metabolism than exposure to NRCWE-061 (MWCNTs-NH_2_) and NRCWE-062 (MWCNTs-pristine) (Figure 6A). Some of these pathways were found to be activated (Figure 6B). 

At 54 µg/mouse, all CNTs except for NM-411 (SWCNTs-pristine) perturbed canonical pathways; some similarities and differences were found among the top 15 significant canonical pathways organized by *p*-value or z-score (Figure 7). At this high dose, NRCWE-051 (SWCNTs-pristine) most significantly perturbed several pathways, such as inflammation, the metabolism of xenobiotics and lipids, and coagulation (Figure 7A), which were found to be activated, and the metabolism of xenobiotics pathway was found to be inhibited (Figure 7B). NRCWE-055 (SWCNTs-pristine) and NRCWE-057 (SWCNTs-COOH) perturbed more canonical pathways than NRCWE-056 (SWCNTs-OH) including pathways related to inflammation and the inactivation of the antioxidant vitamin C. NRCWE-056 (SWCNTs-OH), NRCWE-052 (SWCNTs-pristine), NRCWE-053 (SWCNTs-OH), NRCWE-054 (SWCNTs-COOH) and NRCWE-061 (MWCNTs-NH_2_) activated the cell cycle control of chromosomal replication and the NER pathway (Figure 7B). All SWCNTs regulated pathways related to inflammation (Figure 7). 

Pathways related to inflammation, lipid and xenobiotic metabolism were regulated with respect to the MWCNTs, but agranulocyte/granulocyte adhesion and diapedesis were over-represented for all MWCNTs (Figure 7A). NRCWE-061 (MWCNTs-NH_2_) regulated more pathways than the rest of the MWCNTs, with xenobiotic metabolism pathways inhibited (Figure 7B). All MWCNTs inhibited the erythropoietin signaling pathway (Figure 7B). 

Of the many perturbed canonical pathways, acute-phase response signaling, agranulocyte adhesion and diapedesis, granulocyte adhesion and diapedesis, IL-10 signaling and LXR/RXR activation, the role of IL-17F in allergic inflammatory airway diseases, and tumor microenvironment pathways were commonly perturbed by all CNTs (Figure 7). Comparisons of DEGs in the commonly perturbed canonical pathways in the 18 and 54 µg/mouse dose groups are presented in Appendix A, respectively, and described in the Appendix A.

### 3.5. WGCNA Dynamic Modules and GSEA

IPA analysis allowed us to determine the perturbed canonical pathways based on DEGs after exposure to different CNTs. However, we considered WGCNA/GSEA a second approach to determine if the 12 CNTs regulated specific pulmonary toxicity-related modules. From 43 publicly available transcriptomic studies, WGCNA was used to identify 23 gene modules (Appendix A). After the enrichment analysis using DAVID bioinformatics and IPA, the GO biological process, GO molecular functions, KEGG pathways, IPA canonical pathways, upstream regulators, diseases, and disorders related to each module were identified and are presented in Appendix A. Some specific modules could be grouped based on the similarities of their canonical pathways identified with the IPA analysis (Appendix A). The canonical pathways enriched in modules 2 and 13 were related to apoptosis signaling, death receptor signaling, ATM signaling, senescence, p53 signaling, Myc-mediated apoptosis, and PI3/AKT signaling. Modules 5 and 6 shared canonical pathways related to DNA damage, such as DNA double-strand break repair by non-homologous end joining, kinetochore metaphase signaling, the cell cycle control of chromosomal replication, and the cell cycle G2/M DNA damage checkpoint regulation. Modules 7 and 19 consisted of pathways associated with mitochondrial dysfunction, oxidative phosphorylation, and sirtuin signaling. Modules 8 and 16 were associated with aldosterone signaling epithelial cells, glutathione-mediated detoxification, and unfolded protein response (Appendix A). 

Inflammation pathways were represented in eight modules (9, 10, 20, 11, 14, 18, 21, and 15), with some notable differences in the specific types of inflammatory pathways. For example, module 9 was enriched in phagosome maturation, phagosome formation, and the role of MAPK signaling in promoting the pathogenesis of influenza, among others. Module 10 was enriched in granulocyte/agranulocyte adhesion and diapedesis, acute-phase response signaling, and IL-10 signaling. Interferon signaling, the role of hypercytokinemia/hyperchemokinemia, and the antigen presentation pathway were some of the most enriched pathways in module 20. Module 11 was enriched in acute-phase response signaling, the tumor microenvironment pathway, and IL-6 signaling. Modules 14, 18, and 21 were enriched in the Th1 and Th2 activation pathways. Module 15 was enriched in TWEAK signaling, the production of nitric oxide and reactive oxygen species in macrophages, and TNFR1 and TNFR2 signaling (Appendix A). 

Module 23 was mainly associated with FXR/RXR and LXR/RXR activation (Appendix A). Modules 4, 22, 17, and 12 shared pathways associated with calcium signaling, circadian rhythm, breast cancer regulation by stathmin1, and cardiomyopathy signaling. Other non-specific modules were module 1 (enriched in cardiac hypertrophy signaling, axonal guidance signaling, the hepatic fibrosis signaling pathway, the molecular mechanism of cancer, and thrombin signaling) and module 3 (the FAT10 signaling pathway, polyamine regulation in colon cancer, and the BAG2 signaling pathway) (Appendix A).

GSEA was used to identify which of those modules were perturbed after exposure to CNTs. Comparisons of all CNTs for each dose are presented in Figure 8, Figure 9 and Figure 10. 

Almost all modules were significantly enriched at the low dose of 6 μg/mouse for NRCWE-051 (SWCNTs-pristine) and NRCWE-061 (MWCNTs-NH_2_). Apoptosis, death receptor, and ATM signaling (module 2) had positive NES values, indicating enrichment at the top of the ranked list in NRCWE-063 (MWCNTs-OH) (Figure 8). For the SWCNTs, the modules related to phagosome maturation, granulocyte adhesion, and diapedesis (modules 9 and 10) showed positive NES values, but other modules related to late inflammatory responses were enriched at the bottom of the ranked list (modules 20, 14 and 18) (Figure 8). In contrast, modules associated with the Th1 and Th2 activation pathways (modules 14 and 18) had positive NES values in all groups of MWCNTs. Modules related to mitochondrial dysfunction, oxidative phosphorylation, the unfolded protein response, calcium signaling and epithelial adherens junction signaling (modules 7, 19, 8, 21, and 12) had negative NES values in MWCNTs, except for NRCWE-061 (MWCNT-NH_2_) (Figure 8).

At the medium dose of 18 μg/mouse, more modules had positive NES values and some differences were observed across the CNT types. Modules related to tRNA charging, the sirtuin signaling pathway, and the cell cycle (modules 5 and 6) showed positive NES values across all CNTs, except for NRCWE-063 (MWCNTs-OH). Most of the modules related to inflammation were associated with exposure to CNTs in general, except for modules associated with the Th1 and Th2 activation pathways (14 and 18), which were only found enriched at the top of the ranked list in the MWCNT groups. Modules 1, 3, 4, 22, 17, and 12 did not show specific patterns among the CNT types (Figure 9).

At the highest dose of 54 μg/mouse, many modules had positive NES values across all CNTs. For example, modules 5 and 6 (tRNA charging, the sirtuin signaling pathway, and the cell cycle) and module 16 (the unfolded protein response and cellular stress) were significantly perturbed. Modules 9, 10, 20, 11, 21, and 15 (related to inflammation such as inflammatory signaling pathways) were over-represented at the top of the ranked list across all CNTs. Module 3, related to FAT10 and BAG2 signaling pathways, had positive NES values in almost all CNTs, except for NM-411 (SWCNTs-pristine). At 54 μg/mouse, modules 14 and 18 (Th1 and Th2 pathways) were more enriched in the MWCNTs than the SWCNTs. In contrast, modules 2, 13, and 8 (related to apoptosis and oxidative stress) were under-represented at the bottom of the ranked list in almost all CNTs (Figure 10).

Figure 11 shows the correlation between the NES of each module and the total deposited surface area (BETxDose). Only eight modules were significantly correlated. A strong negative correlation (r = −0.7473) was found between module 13 (apoptosis signaling and death receptor signaling) and the total deposited surface area. Some modules were strongly correlated (r = 0.4269–0.6801) with the total deposited surface area, such as modules 16 (aldosterone signaling epithelial cells, glutathione-mediated detoxification, and the unfolded protein response); 5 (DNA damage); and 9, 10, 15 and 20 (inflammation). Module 6 (cell cycle control) showed a moderate correlation (r = 0.3633) with the total instilled surface area. The top five canonical pathways and upstream regulators for each module are also shown in Figure 11.

### 3.6. Pro-Fibrotic Transcriptomic Signature

A panel of 17 pro-fibrotic genes (PFS17) was identified and validated in vivo to classify substances as pro-fibrotic or non-fibrotic [14]. Figure 12, Appendix A show heatmaps that compare the transcriptomic profiles of lung-specific diseases versus the PFS17 of lungs exposed to the SWCNTs and MWCNTs. NRCWE-051 (SWCNTs-pristine), at 54 μg/mouse, was the only CNT predicted as pro-fibrotic (Figure 12). Appendix A shows the results of the PCA performed using the prcomp function in R using the dataset of publicly available in vivo lung transcriptomic studies described by Rahman et al. [14] (Appendix A) and the dataset produced in this study. The scatter plot of the first two components is presented. The red line reflects the contrast between the pro-fibrotic and non-fibrotic classes, and the grey line separates borderline pro-fibrotic profile with their first principal component being negative. Among all tested CNTs, NRCWE-051 (SWCNTs-pristine) at 54 μg/mouse was the only CNT identified in the pro-fibrotic class (Appendix A). 

### 3.7. Potency Ranking of CNTs

Appendix A shows the results from the BMD analysis, which ranked NRCWE-051 (SWCNTs-pristine) as highly potent, showing the lowest BMD, and NM-411 (SWCNTs-pristine) as the least potent. The BMD dose–response modelling and potency estimation revealed that only NRCWE-051 (SWCNTs-pristine) showed BMD < lowest dose (6.71 × 10^−5^(Reactome pathway), 4.77 × 10^−8^ (25th ranked gene) and 1.06 × 10^−12^ (LCRD) μg/mouse for NRCWE-051 (SWCNTs-pristine). The rest of the CNTs, except for NM-411, showed BMD estimates spanning from 4.5 to 19 μg/mouse based on the Reactome pathway and the 25th gene and from 2.3 to 6.01 μg/mouse based on the LCRD. NM-411 showed the highest values of ~26 μg/mouse (Reactome pathway, 25th gene) and 6.16 μg/mouse (LCRD).

## 4. Discussion

In the present study, the pulmonary toxicity of SWCNTs and MWCNTs with different types of surface functionalization was evaluated in mice following the single intratracheal instillation of 6, 18, or 54 μg/mouse doses of individual SWCNTs and MWCNTs. BALF cellularity was assessed as an indicator of initiation of lung inflammation, and DNA damage was assessed to determine the genotoxic potential of CNTs. The global transcriptomic responses in lungs were evaluated to identify the underlying mechanisms of toxicity, and the resulting data were used to rank the potency of CNTs to induce pathway perturbations. The selected doses of 6, 18, and 54 μg/mouse approximated 1/3, 1, and 3 times, respectively, the occupational exposure at the exposure limit of 1 μg carbon/m^3^ for 40 years assuming a 40 h working week, as recommended by NIOSH [4,12]. 

### 4.1. Genotoxic Effects of CNTs

Only a few CNTs elicited genotoxic effects. The evaluation of DNA strand breaks in BALF cells after 1 and 28 days of exposure revealed that only three individual MWCNTs, NRCWE-062 (MWCNTs-pristine), NRCWE-063 (MWCNTs-OH), and NRCWE-064 (MWCNTs-COOH), induced DNA damage compared with the control with ~6–12% DNA in the tail on post-exposure day 1 at 6 and 18 μg/mouse and ~5–9% DNA in the tail on post-exposure day 28 at 6, 18, or 54 μg/mouse.

MWCNTs did not induce DNA damage in lung tissues. The evaluation of DNA strand breaks in lung tissues revealed that NRCWE-053 (SWCNTs-OH) and NRCWE-054 (SWCNTs-COOH) induced a slight increase in DNA breaks (~5–12%) on days 1 and 28 post-exposure to 6 or 18 μg/mouse of CNTs. Lung gene expression profiles showed activated pathways associated with DNA damage (the cell cycle control of chromosomal replication and the nucleotide excision repair pathway) on day 1 post-exposure to NRCWE-052 (SWCNTs-pristine), NRCWE-053 (SWCNTs-OH), NRCWE-054 (SWCNTs-COOH), NRCWE-056 (SWCNTs-OH), NRCWE-061 (MWCNTs-NH_2_), and NRCWE-062 (MWCNTs-pristine) at 54 μg/mouse, which was consistent with the subtle increase in DNA strand breaks observed in lung tissue for NRCWE-053 (SWCNTs-OH) and NRCWE-054 (SWCNTs-COOH). These results suggest that pristine MWCNTs and functionalized MWCNTs are capable of inflicting genotoxicity in BALF cells but not at the tissue level at the doses and post-exposure time points investigated, whereas functionalized SWCNTs are potentially acutely DNA-damaging to lung tissue after exposure.

These results are not consistent with what has been reported in the literature about MWCNTs’ potential to induce tissue genotoxicity and alter pathways associated with DNA damage. In a study involving the occupational exposure of workers to MWCNTs, the analysis of ncRNA and mRNA expression profiles in the whole blood samples of workers exposed to MWCNTs aerosols in a manufacturing facility showed altered gene networks related to cell cycle regulation and progression, apoptosis, cell proliferation, and carcinogenic pathways at the average inhalable elemental carbon concentration of ~14.42 ± 3.8 μg/m^3^ compared with unexposed workers. The study highlighted the potential of MWCNTs to induce pulmonary toxicity and carcinogenicity in humans [82].

Other studies have found pathways associated with lung inflammation, cytotoxicity, cellular stress, and genotoxicity after exposure to high doses of MWCNTs in lung tissues. The biological processes associated with cell death, cell proliferation, and free radical scavenging were perturbed in mice 90 days after their first exposure to 36 or 109 μg/mouse of Mitsui-7 or 78 μg/mouse of NM-401 (both MWCNTs) once a week for four consecutive weeks [9]. In another study, the toxicity of short and long MWCNTs at 18, 54, and 162 μg/mouse was associated with the differential expression of genes related to the immune response, cell cycle, and response to wounding pathways on day 1 post-exposure [10]. 

The genotoxicity of MWCNTs with different diameters and functional groups was also evaluated at 6, 18, or 54 μg/mouse. DNA damage was observed in mouse lungs after 1 and 28 days of exposure to thick and functionalized MWCNTs. Thick hydroxylated or thick carboxylated MWCNTs induced a higher DNA damage response compared with thick pristine MWCNTs. In the same study, increased DNA strand breaks were observed in BALF cells after exposure to thick or thin MWCNTs, showing the following genotoxic trend of MWCNTs-COOH > MWCNTs-OH > MWCNTs-pristine. Despite this tendency, DNA strand breaks in lung tissue were only found to positively correlate with MWCNT diameter [12]. Lastly, seven different MWCNTs (long and thick NRCWE-006/Mitsui-7 and NM-401, short and thin NM-400 and NM-403, pristine NRCWE-040, hydroxylated NRCWE-041, and carboxylated NRCWE-042) showed a differential genotoxic potential that was correlated with in vitro DNA damage in the macrophage THP-1 cell line [83]. To date, only one type of MWCNT (Mitsui-7, pristine, long, straight, and rigid MWCNTs) has been classified as possibly carcinogenic to humans, which cannot be generalized to other types of CNTs based on available data. There is also evidence that long and thick, short, thin and entangled, and even double-walled carbon nanotubes can induce adenocarcinoma and carcinoma following pulmonary exposure in rats [84,85,86]. Thus, MWCNTs have been shown to induce genotoxicity and carcinogenicity to variable degrees in different models regardless of their length, diameter and surface properties. 

Not many studies have systematically investigated the genotoxicity of SWCNTs. In male F344 rats exposed to 0.2 or 1 mg/kg of SWCNTs (long or short) via a single intratracheal instillation, no significant increase in % DNA in the tail was observed in lung tissue 26 days after exposure [87] or in the lungs of Crl:CD(SD) rats after exposure to a single intratracheal instillation of 0.2 or 1 mg/kg or the repeated instillation of 0.2 mg/kg once a week for 5 weeks [88]. In contrast, the intratracheal instillation of 54 μg/mouse of SWCNTs induced an increase in % DNA in the tail after 3 h of exposure in C57BL/6 Apoetm1 (ApoE-/-) mice [89]. 

In the present study, in addition to quantifying DNA strand breaks and assessing gene expression profiles, WGCNA/GSEA analysis was conducted to identify a group of genes and processes that are specific for pulmonary toxicity. The results showed that modules associated with DNA damage (modules 5 and 6) were significantly enriched at the top of the ranked list in a dose-dependent manner across all CNTs, and both modules positively correlated with the total deposited surface area, which indicates that exposure to SWCNTs and MWCNTs may result in impaired cell cycle processes and lead to genetic instability and genotoxicity after long-term repeated exposure or exposure to higher doses than 54 μg/mouse, which represents 3 times the occupational exposure at the exposure limit of 1 μg carbon/m^3^ for 40 years, assuming a 40 h working week, as recommended by NIOSH [4].

The different responses observed between BALF cells and lung tissue is an important finding that confirms the key role of macrophages in the response to nanomaterials exposure [90]. Unlike the uniform tissue distribution of soluble chemicals, following exposures to particles, nanomaterials, and CNTs, only a fraction of the cells, including BALF cells in the lung, will be physically exposed to materials. The activation of macrophages can produce reactive oxygen and nitrogen species, which could lead to cell death and DNA damage [16].

Though the present study showed subtle DNA damage in BALF and in lung tissues post-exposure, the results did not show time, dose, or property dependency, suggesting that more work is needed to understand the mechanisms of action underlying the observed genotoxicity for this class of nanomaterials.

### 4.2. Pro-Inflammatory Effects of CNTs

Lung inflammation is a widely studied and reported tissue endpoint for CNTs. Inflammation and the enrichment of DEGs associated with the inflammation process were observed across all CNTs in the present study; however, the type of inflammatory pathways activated or inhibited varied depending on the dose and type of tested CNTs. 

It has been well-described that neutrophils are the first line of host defense against infection by pathogens. Although all SWCNTs in this study were capable of inducing neutrophil influx, the results indicated that NM-411 (SWCNTs-pristine), NRCWE-055 (SWCNTs-pristine), and NRCWE-056 (SWCNTs-OH) induced the smallest response, whereas NRCWE-051 (SWCNTs-pristine) was the most inflammogenic, triggered the recruitment of neutrophils to lungs 1 day after exposure to the lowest dose of 6 μg/mouse, and induced inflammation for up to 28 days at 18 and 54 μg/mouse. The pro-inflammatory response of NRCWE-051 (SWCNTs-pristine) was supported by several canonical pathways perturbed at the lowest dose of 6 μg/mouse, associated with the activation and recruitment of neutrophils (IL-17F, granulocyte adhesion, and diapedesis), agranular leukocytes (agranulocyte adhesion and diapedesis), the induction of cytokine dysregulation, and the activation of the acute-phase response. In addition to inflammatory pathways, NRCWE-051 (SWCNTs-pristine) perturbed pathways that promote cell survival, adaptation, and the tumor microenvironment after 1 day of exposure at 6 μg/mouse. 

Among the MWCNTs, NRCWE-061 (MWCNTs-NH_2_) induced persistent inflammation for up to 28 days post-exposure at both 18 and 54 μg/mouse. Although the IPA analysis did not identify significant canonical pathways at 6 μg/mouse for NRCWE-061 (MWCNTs-NH_2_), the WGCNA/GSEA analysis revealed that this CNT type shared some modules with NRCWE-051 (SWCNTs-pristine) at the three doses. For example, modules 5, 6, 9 and 10 (associated with the unfolded protein response, phagosome maturation, agranulocyte/granulocyte adhesion, and the acute-phase response) were common to both types of CNTs (NRCWE-051 and NRCWE-061). It has been reported that pristine MWCNTs more readily induce fibrosis than functionalized MWCNTs in vitro and in vivo [15], whereas an aminated surface is suggested to increase toxicity because of its augmented ability to interact with the cell membrane and organelles [91]. 

At the medium dose of 18 μg/mouse, IPA and WGCNA/GSEA analysis showed that NRCWE-051 (SWCNTs-pristine), NRCWE-061 (MWCNTs-NH_2_), NRCWE-054 (SWCNTs-COOH) and NRCWE-062 (MWCNTs-pristine) significantly perturbed pathways associated with inflammation. However, NRCWE-051 (SWCNTs-pristine) perturbed a larger number of inflammatory pathways and showed a dose-dependent transition from inflammation at the low dose to cellular stress at the highest dose. NRCWE-054 (SWCNTs-COOH) showed similar responses to NRCWE-051 (SWCNTs-pristine), though with a lesser magnitude and with a difference in the inactivation of xenobiotic metabolism pathways, which were only activated after exposure to NRCWE-054 (SWCNTs-COOH). 

NRCWE-061 (MWCNTs-NH_2_) and NRCWE-062 (MWCNTs-pristine) also perturbed various inflammatory IPA canonical pathways including the acute phase, agranulocyte/granulocyte, and IL-17 signaling pathways. Moreover, WGCNA/GSEA analysis identified the Th1 and Th2 activation pathways (modules 14 and 18) as enriched at the top of the ranked list in all MWCNT groups and enriched at the bottom of the ranked list in all SWCNT groups. These results suggest that the MWCNTs induced the upregulation of genes involved in the Th1 and Th2 pathways, whereas the SWCNTs, except for NRCWE-051 (SWCNT-pristine), downregulated genes related to the Th1 and Th2 responses. Specifically, most SWCNTs induced more downregulated genes related to the Th1 response than MWCNTs at the low dose. The Th1 immune response produces pro-inflammatory cytokines, stimulates phagocytosis, and shows antifibrotic activity [92,93]. In contrast, the Th2 immune response produce type 2 cytokines and inflammatory mediators, a part of the adaptive immune response and resolution phase of inflammation. The Th2 response is also induced in the wound healing process and is implicated in lung fibrosis induced by CNTs [94,95]. 

The transition from an acute inflammatory response to chronic fibrosis through Th1 and Th2 signaling was described in a study where MWCNTs induced fibrosis in C57BL/6 mice exposed to 40 μg/mouse after 14 days of exposure [96]. In another study, the role of crucial mediators of the Th2 response, signal transducer and activator of transcription 6 (STAT6), and interleukin-1 receptor 1 (IL-1R1) in MWCNT-induced lung inflammation and fibrosis was investigated in wild-type C57BL/6, IL-1R1 knockout (KO) or STAT6 (KO) mice exposed to 162 μg/mouse of Mitsui-7 via intratracheal administration. These results demonstrated that IL-1R1 deficiency suppressed acute inflammation but did not inhibit the eventual pro-fibrotic response, whereas exposed STAT6 KO mice showed attenuated acute inflammation and pro-fibrotic response up to 28 days post-exposure to MWCNTs compared with wild-type mice [97]. Perturbed pathways associated with the Th2-M2 response were also identified in the gene expression profiles from the lungs of C57BL/6 mice exposed to different doses of pristine MWCNTs (NM-401, NRCWE-26, and Mitsui-7) [98]. Pristine MWCNTs have been shown to induce fibrosis compared with functionalized MWCNTs in vitro and in vivo [15]. 

These results collectively suggest that at the medium dose of 18 μg/mouse, the potency to induce an inflammatory response follows a trend: NRCWE-051 (SWCNTs-pristine) > NRCWE-054 (SWCNTs-COOH), NRCWE-061 (MWCNTs-NH_2_) and NRCWE-062 (MWCNTs-pristine) > the rest of CNTs.

At the highest dose of 54 μg/mouse, most of the canonical pathways and modules were over-represented across all CNTs, indicating that all CNTs triggered a similar biological response at the high dose. Inflammatory pathways were significantly perturbed and included acute-phase response signaling, agranulocyte/granulocyte adhesion and diapedesis, IL-10 signaling, and LXR/RXR activation. Moreover, acute-phase response signaling, the role of IL-17F in allergic inflammatory airway diseases, and the tumor microenvironment were activated for all CNTs. These canonical pathways were also reported to be perturbed after exposure to 162 μg/mouse of MWCNTs in C57BL/6 mice after 1 and 3 days following MWCNT exposure [10]. When comparing the results from the present study with the results reported by others, it must be pointed out that inflammatory modules also correlated with the total deposited surface area [12].

The above indicates that the recruitment of leukocytes to the site of lung tissue damage, the secretion of pro- and anti-inflammatory cytokines, and the diffusion of those mediators leading to a systemic acute-phase reaction [99] are triggered for CNTs regardless of their physico-chemical properties at 54 μg/mouse. Some of the DEGs implicated in those common canonical pathways are *Saa1* (serum amyloid A1), *Saa3* (serum amyloid A3), *IL-6* (interleukin 6), *C3* (complement component C3), and *Serpin* (serine protease inhibitors) in acute-phase response signaling; *Cxcl6* (C-X-C motif chemokine ligand 6), *Ccl7* (C-C motif chemokine ligand 7), *Ccl17* (C-C motif chemokine ligand 17), *Cxcl10* (C-X-C motif chemokine ligand 10), *Ccl8* (C-C motif chemokine ligand 8), *Cxcl3* (C-X-C motif chemokine ligand 3), and *Cxcr1* (C-X-C motif chemokine receptor 1) in agranulocyte/granulocyte adhesion and diapedesis; *Lcn2* (lipocalin 2) involved in IL-17 signaling; *Saa1*, *C3*, *Cd14* (monocyte differentiation antigen Cd14), *Itih4* (inter-alpha-trypsin inhibitor heavy chain 4), *Il-33* (interleukin 33), and *Il-6* in LXR/RXR activation.

Despite the similarities observed at the pathway level, there were also notable differences at the gene level. In this study, the transcription of *Saa1* increased after exposure to SWCNTs (3.6–14.8 fold-change) and MWCNTs (2.2–50 fold-change), whereas the levels of *Saa3* only increased after exposure to MWCNTs (6–120 fold-change) in lung tissue. SAA proteins participate in the acute-phase response and could be induced 1000-fold over a control at 24–36 h after stimulation [100]. Mice express *Saa1*, *Saa2*, and *Saa3*, whereas humans express *SAA1* and *SAA2*. SAA proteins are found induced in patients with lung cancer, sarcoidosis, idiopathic pulmonary fibrosis, and chronic obstructive pulmonary disease [101], so they may be used as a potential clinical biomarker of different lung diseases. The role of acute-phase proteins in cardiovascular diseases is well-established [102].

Many studies have reported the increased expression of *Saa* mRNA and SAA protein levels in lung tissue, BALF, and plasma after exposure to nanomaterials [99,102,103]. For example, *Saa3* mRNA expression was evaluated in the lung tissue of C57BL/6J mice exposed to MWCNTs, SWCNTs, and other nanomaterials via the inhalation or intratracheal instillation at different doses and time points up to 4 weeks post-exposure. For TiO_2_, carbon black nanoparticles, and CNTs, *Saa3* mRNA and SAA3 protein levels in BALF and plasma increased after 1, 3, and 28 days following exposure in mice exposed by intratracheal instillation to 18, 54, and 162 μg/mouse and correlated with the number of neutrophils in BALF [104]. In C57BL/6 mice exposed via intratracheal instillation, a 297-fold induction of *Saa3* was observed after exposure to short MWCNTs on day 3 post-exposure at 54 μg/mouse and a 184-fold expression was observed after exposure to 162 μg/mouse of long MWCNTs [10].

*Saa3* induction was also observed on day 1 post-instillation in mouse lungs exposed to different types of TiO_2_ NPs at 18, 54, and 162 μg/mouse NPs [105,106]; 0.7 or 2 μg/mouse of uncoated and coated ZnO NPs [107]; and increased *Saa1* and *Saa3* mRNA levels in C57BL/6BomTac mouse lungs after 5 days following exposure to 42.4 ± 2.9 mg/m^3^ of coated TiO_2_ NPs for 11 consecutive days via whole-body inhalation [108]. In some of these studies, the levels of *Saa* mRNA expression correlated with the instilled surface area of nanomaterials [106]. 

Moreover, the pathway of cholesterol biosynthesis related to lipid metabolism was perturbed in mice exposed to NRCWE-051 (SWCNTs-pristine), and the LXR/RXR activation pathway was perturbed after exposure to all CNTs. In recent years, the impairment of pulmonary lipid homeostasis has attracted attention from toxicologists, as it is essential for gas exchange, regulates innate immune responses, and has been extensively discussed in studies investigating the effects of cigarette smoke and cigarette vapors [109,110]. In the context of nanomaterials, the destruction of pulmonary surfactant function related to lipid homeostasis and the damage of lamellar bodies has been documented [111,112,113]. The role of lipid mediators in tissue fibrosis and other chronic pulmonary diseases was reviewed by Agudelo et al. [114], highlighting the necessity of more studies to elucidate the mechanisms of nanomaterials on lipid metabolism in lung diseases.

Some other important pathway perturbations include the upregulation of rhythmic and circadian rhythm processes at medium and high doses after exposure to NRCWE-063 (MWCNTs-OH) and NRCWE-064 (MWCNTs-COOH) with the differential expression of several associated genes including *Dbp* (albumin D site-binding protein), *Per2* (period circadian clock 2), *Per3* (period circadian clock 3), *Hlf* (hepatic leukemia factor), *Tef* (thyrotroph embryonic factor), *Bhlhe40* (basic helix–loop–helix family, member e40), *Nr1d1* (nuclear receptor subfamily 1, group D, member 1), and *Nr1d2* (nuclear receptor subfamily 1, group D, member 2). Approximately 10% of the transcriptome, including the extracellular matrix, is under circadian control, and lung tissue exhibits strong circadian rhythms because of its high elastin and collagen content [115]. It has been described that the circadian clock regulates the activation of the NRF2/glutathione-mediated antioxidant pathway to modulate pulmonary fibrosis [116], and the expression of PER2 (a clock protein) in bronchiolar epithelial cells is responsive to glucocorticoids [117]. Altered molecular clocks were observed following cigarette smoke and LPS (lipopolysaccharides) exposures, bacterial/viral infections in mice and in patients with chronic airway diseases [118]. However, more studies are needed to elucidate the effect of MWCNT-induced changes in circadian rhythm and potential physiological implications in the lung tissue. 

### 4.3. Pro-Fibrotic Effects of CNTs

According to Adverse Outcome Pathway 173 (AOP173) [119], “Substance interaction with the lung resident cell membrane components leading to lung fibrosis” (https://aopwiki.org/aops/173 (accessed on 2 August 2022), the interaction of a substance with cells can release signals that could lead to the increased secretion of pro-inflammatory cytokines and result in the recruitment of pro-inflammatory cells to the site of damage. Persistent inflammation will lead to tissue injury and subsequently to a wound healing response involving the Th2/M2 response and the secretion of interleukins and growth factors, leading to fibroblast and myofibroblast proliferation. Prolonged tissue injury and impaired healing process results in excessive extracellular matrix deposition, leading to tissue fibrosis. AOP173 has been used to identify key events linking pristine MWCNT exposure to lung fibrosis [98] and to develop a pro-fibrotic gene signature to predict the occurrence of lung fibrosis [14]. In a previous study, Rahman et al. [14] developed a PFS17 that is specifically predictive of lung fibrosis in mice. Among all CNTs and various experimental conditions tested in the present study, the 54 μg/mouse dose of NRCWE-051 (SWCNTs-pristine) was the only experimental condition identified in the pro-fibrotic class by the 17-gene pro-fibrotic biomarker panel. This result provides evidence that PFS17 is a promising biomarker panel that can be used to evaluate the safety of different CNTs.

Moreover, the lung is mechanically stressed during the respiratory cycle, and the extracellular matrix is involved in signaling pathways associated with cell growth, proliferation and gene expression. When there is an injury or external stimuli that disrupts the mechanical homeostasis, many signaling pathways associated with repair, such as pathways associated with integrins, growth factor receptors, G-protein couple receptors, mechanoresponsive ion channels, and cytoskeletal signaling, are triggered [120,121]. Although specific canonical pathways implicated in the mechanotransduction of the lung were not identified, module 12 (which consisted of related pathways such as calcium signaling, epithelial adherens junction signaling, and hepatic fibrosis) were over-represented following exposure to NRCWE-051 (SWCNTs-pristine) at the three doses. 

Studies have reported that the induction of fibrosis could be associated with the fiber length of SWCNTs, as was observed in mice exposed to 40 μg/mouse of MWCNTs on day 90 post-exposure [122]; in contrast, the functionalization seemed to decrease the extent of fibrosis induced by MWCNTs in C57BL/6 mice exposed to 18, 54 and 168 μg/mouse after 1 and 28 days of exposure [14] and after 21 days of exposure at 2 mg/kg [15]. The toxicity observed for NRCWE-051 (SWCNTs-pristine) cannot be attributed to the fiber length, as the information provided by the manufacturer could not be confirmed. Thus, the pristine surface of the NRCWE-051 (SWCNTs-pristine) seems to have influenced the pro-fibrotic response. 

BMD analysis ranked NRCWE-051 (SWCNTs-pristine) as the most potent CNT and NM-411 (SWCNTs-pristine) as the least potent from this panel. It is important to note that NRCWE-051 (SWCNTs-pristine) perturbed biological functions at 6 μg/mouse as early as 1 day after exposure. According to Poulsen et al. [12], this dose approximates a third of the expected 40-year exposure for workers at the exposure limit of 1 μg carbon/m^3^ recommended by NIOSH [4] assuming 10% deposition, a ventilation rate of 1.81l/h for mice, and a 40-hour working week. This means that actions need to be taken to minimize exposure to CNTs below the present exposure limit. Although more studies are needed to clearly identify the potential of this type of SWCNT to induce toxicity, the human-equivalent dose at which the response is observed can potentially be used to guide future studies looking for early biomarkers.

## 5. Conclusions

This comprehensive toxicity profile of SWCNTs and MWCNTs with different physico-chemical properties revealed that MWCNTs were more genotoxic than SWCNTs and that all CNTs induced tissue inflammation at lower doses, although the underlying mechanisms differed. At higher doses, the response was exacerbated and perturbation in pathways associated with cellular stress and tissue damage were observed. Of all CNT types, NRCWE-051 (SWCNTs-pristine) was inflammogenic, classified as pro-fibrotic by the PFS17 gene panel, and found to be the most potent, whereas NM-411 (SWCNTs-pristine) was found to be the least potent CNT. Importantly, the biological responses observed at low doses suggest that the exposure limit could be overestimated. The transcriptomic profiles of the twelve CNTs provide information on potential biomarkers and mechanisms of toxicity that can be used for grouping and hazard identification. 

## Figures and Tables

**Figure 1 nanomaterials-13-01059-f001:**
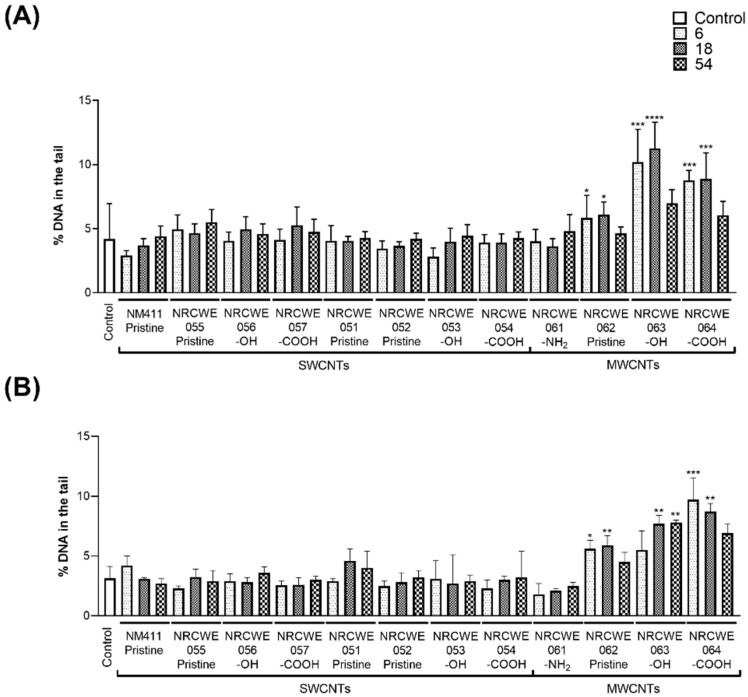
Percentage of DNA in the tail in BALF cells after exposure to individual SWCNTs or MWCNTs. Data represent median values with interquartile range. Statistically significant differences between exposed animals and controls were determined using the Kruskal–Wallis test with Dunn’s post-hoc. * *p* < 0.05, ** *p* < 0.01, *** *p* < 0.001, **** *p* < 0.0001. (**A**) Day 1; (**B**) Day 28. Different colors and fill patterns indicate different dose groups: control, 6 μg/mouse, 18 μg/mouse and 54 μg/mouse of CNTs.

**Figure 2 nanomaterials-13-01059-f002:**
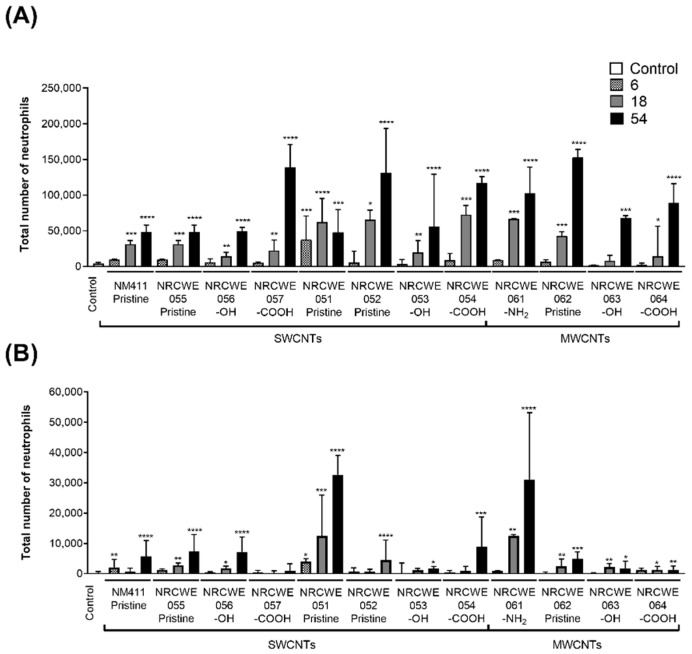
Total number of neutrophils in BALF after exposure to individual SWCNTs or MWCNTs. Data represent median values with interquartile range. Statistically significant differences between exposed animals and controls were determined using the Kruskal–Wallis test with Dunn’s post-hoc. * *p* < 0.05, ** *p* < 0.01, *** *p* < 0.001, **** *p* < 0.0001. (**A**) Day 1; (**B**) Day 28. Different colors indicate different dose groups: control, 6 μg/mouse, 18 μg/mouse and 54 μg/mouse of CNTs.

**Figure 3 nanomaterials-13-01059-f003:**
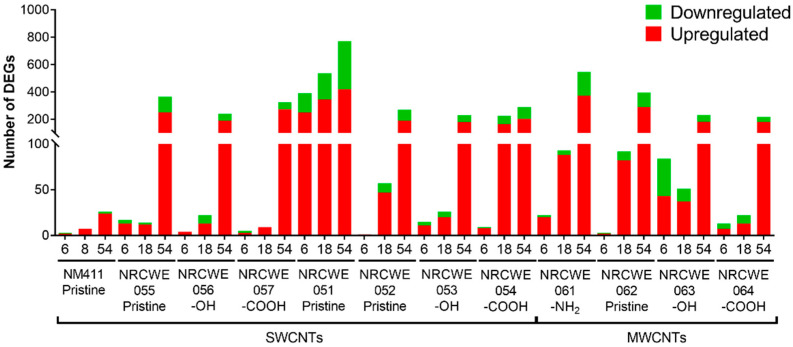
Number of DEGs at 6, 18, and 54 µg/mouse after 1 day of exposure to CNTs. DEGs were defined as genes with an FDR *p*-value ≤ 0.05 and a fold change of ±1.5 in either direction. Red: upregulated. Green: downregulated.

**Figure 4 nanomaterials-13-01059-f004:**
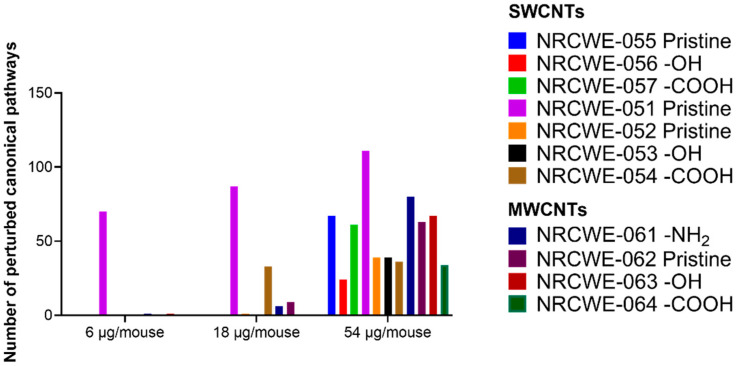
Number of significantly perturbed canonical pathways after exposure to individual SWCNTs or MWCNTs in each dose after 1 day of exposure. The graph represents the number of canonical pathways consisting of at least five DEGs and exhibiting a −log (*p*-value) ≥ 1.3.

**Figure 5 nanomaterials-13-01059-f005:**
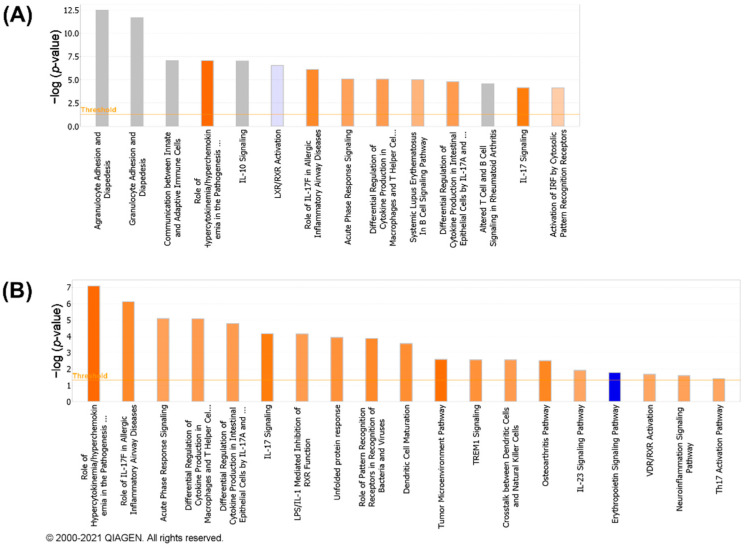
Perturbed canonical pathways after 1 day of exposure to NRCWE-051 (SWCNTs-pristine) at the 6 μg/mouse dose. (**A**) The top 15 canonical pathways were considered significantly perturbed if −log (*p*-value) was ≥ 1.3. (**B**) Pathways were considered significantly activated if −log (*p*-value) was ≥ 1.3 with a z-score ≥ 2 or significantly inhibited if −log (*p*-value) was ≥ 1.3 with a z-score ≤ −2. Blue bars: negative z-score, orange bars: positive z-score, gray bars: no activity pattern available. IPA analysis.

**Figure 6 nanomaterials-13-01059-f006:**
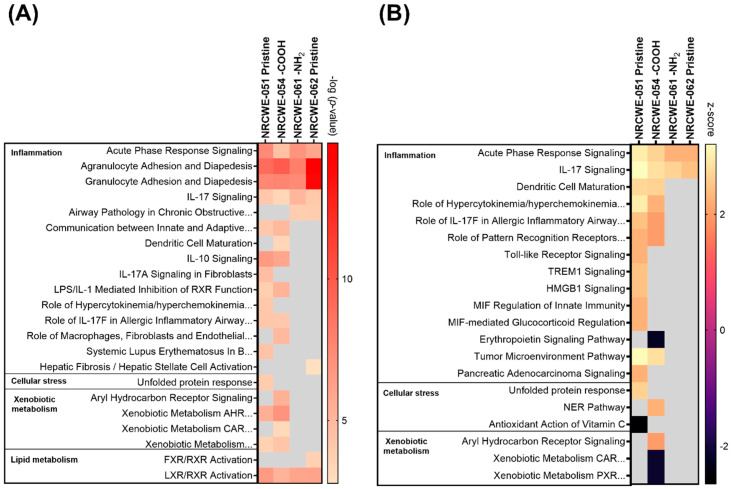
Top 15 perturbed canonical pathways after 1 day of exposure to CNTs at the 18 μg/mouse dose. (**A**) The top 15 canonical pathways were considered significantly perturbed if −log (*p*-value) was ≥ 1.3. (**B**) The top 15 canonical pathways were considered significantly activated if −log (*p*-value) was ≥ 1.3 with a z-score ≥ 2 or significantly inhibited if −log (*p*-value) was ≥ 1.3 with a z-score ≤ −2. Gray represents canonical pathways that were not significant, were not part of the top 15 most significantly perturbed pathways (**A**), or were not significant pathways with highest |z-score| (**B**).

**Figure 7 nanomaterials-13-01059-f007:**
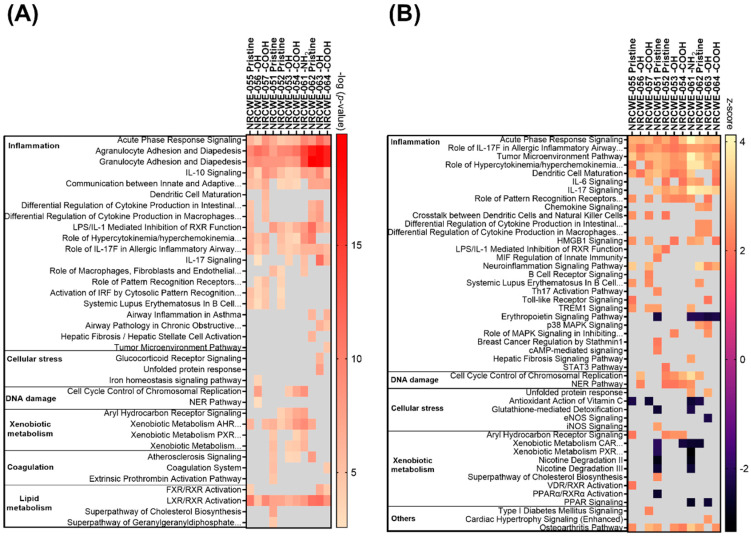
Top 15 perturbed canonical pathways after 1 day of exposure to CNTs at the 54 μg/mouse dose. (**A**) The top 15 canonical pathways were considered significantly perturbed if −log (*p*-value) was ≥ 1.3. (**B**) The top 15 canonical pathways were considered significantly activated if −log (*p*-value) was ≥ 1.3 with a z-score ≥ 2 or significantly inhibited if −log (*p*-value) was ≥ 1.3 with a z-score ≤ −2. Gray represents canonical pathways that were not significant, were not part of the top 15 most significantly perturbed pathways (**A**), or were not significant pathways with highest |z-score| (**B**).

**Figure 8 nanomaterials-13-01059-f008:**
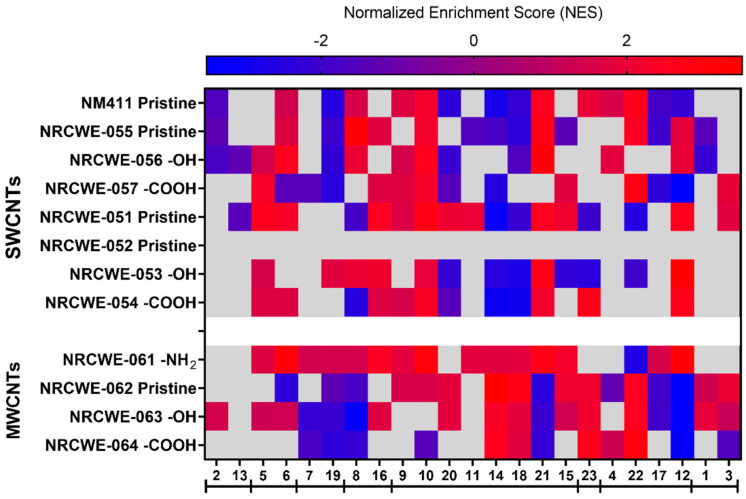
Modules over-represented or under-represented after 1 day of exposure to CNTs at the 6 μg/mouse dose. Red: positive NES values represent gene set enrichment at the top of the ranked list. Blue: negative values represent gene set enrichment at the bottom of the ranked list. Enriched sets were considered significant if their NES had an adjusted *p*-value ≤ 0.05 (gray represents modules that were not significant). The bars group the modules with similar biological functions based on the IPA analysis.

**Figure 9 nanomaterials-13-01059-f009:**
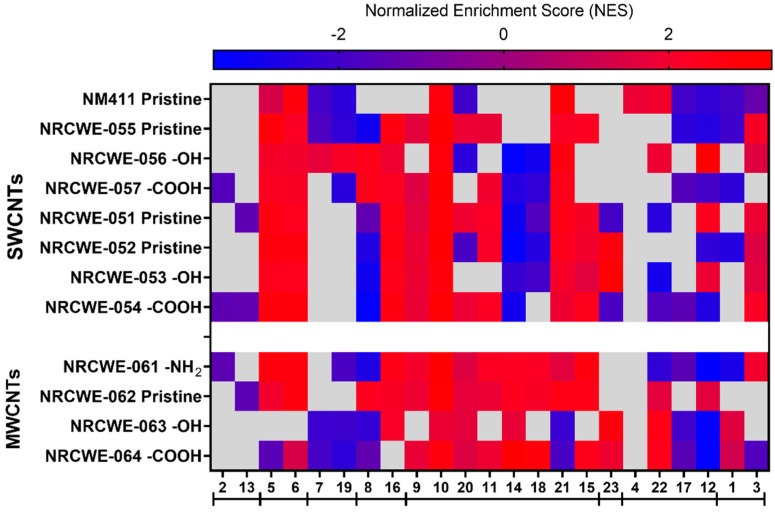
Modules over-represented or under-represented after 1 day of exposure to CNTs at the 18 μg/mouse dose. Red: positive NES values represent gene set enrichment at the top of the ranked list. Blue: negative values represent gene set enrichment at the bottom of the ranked list. Enriched sets were considered significant if their NES had an adjusted *p*-value ≤ 0.05 (gray represents modules that were not significant). The bars group the modules with similar biological functions based on the IPA analysis.

**Figure 10 nanomaterials-13-01059-f010:**
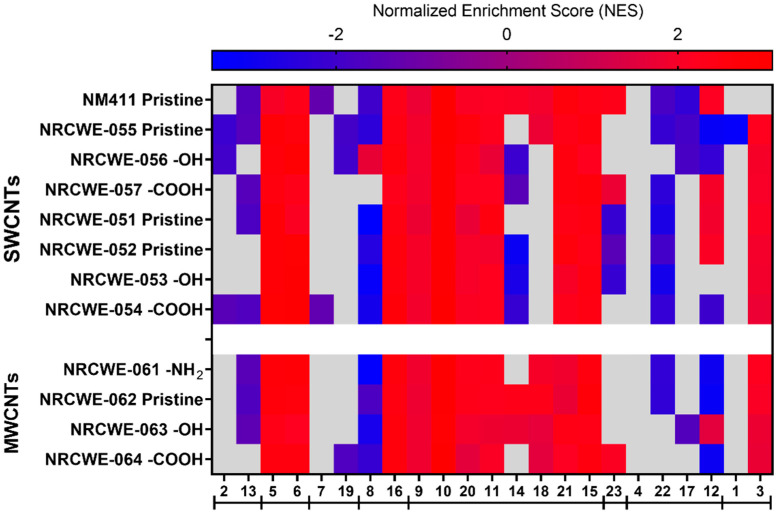
Modules over-represented or under-represented after 1 day of exposure to CNTs at the 54 μg/mouse dose. Red: positive NES values represent gene set enrichment at the top of the ranked list. Blue: negative values represent gene set enrichment at the bottom of the ranked list. Enriched sets were considered significant if their NES had an adjusted *p*-value ≤ 0.05 (gray represents modules that were not significant). The bars group the modules with similar biological functions based on the IPA analysis.

**Figure 11 nanomaterials-13-01059-f011:**
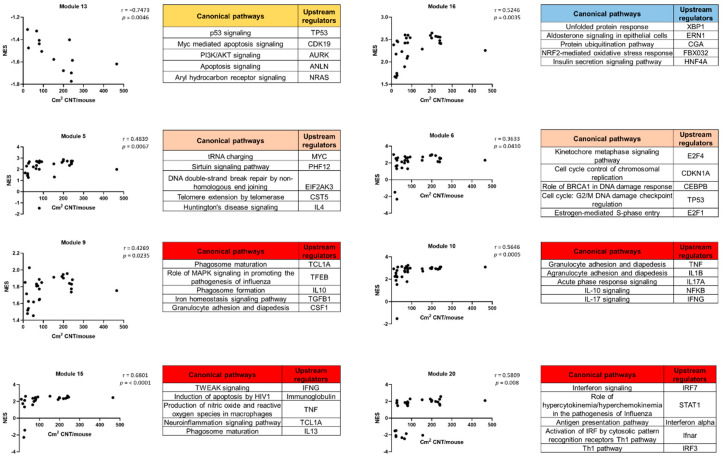
Correlation between the NES of each module and the total deposited surface area (BETxDose) on day 1 post-exposure. Spearman’s correlation was performed with GraphPad Prism 9.2.0 for all CNTs and all doses. The top five canonical pathways and upstream regulators for modules with a significant correlation are shown.

**Figure 12 nanomaterials-13-01059-f012:**
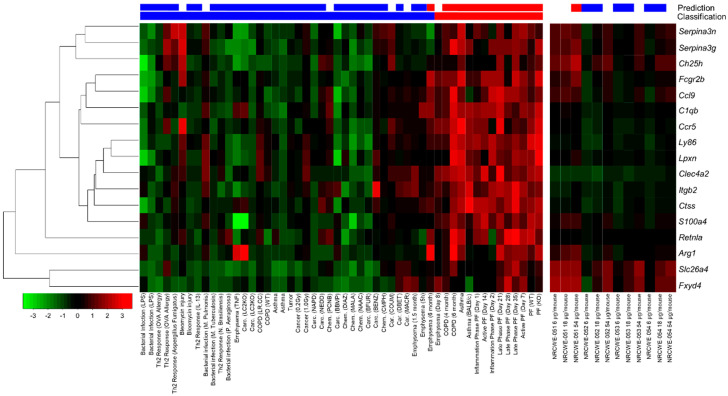
Pro-fibrotic transcriptomic signature after 1 day of exposure to NRCWE-051, NRCWE-052, NRCWE-053, and NRCWE-054. A heatmap that compares the transcriptomic profiles of lung-specific diseases vs. the pro-fibrotic transcriptomic signature (PFS17) [14] of lungs exposed to NRCWE-051, NRCWE-052, NRCWE-053, and NRCWE-054. Classification and prediction results are presented with blue (no-fibrotic), red (pro-fibrotic), or no color (unknown). The bottom bar represents the pre-classification of each study based on the literature data, and the top bar represents outcomes of prediction using PFS17. As described by Williams and Halappanavar et al. [67], bi-clusters related to pulmonary diseases were identified from public microarray datasets. The pro-fibrotic signature was derived from one of the identified bi-clusters. Red: upregulated. Green: downregulated. Appendix A shows the list of pulmonary diseases identified from public microarray datasets [14].

**Table 1 nanomaterials-13-01059-t001:** Physico-chemical properties of CNTs as determined with combustion elemental analysis, BET, SEM and XRF.

Group	Name	Length	Diameter	CNT Type	Carbon	BET	OH	Fe_2_O_3_	CoO	NiO	MgO	MnO
		(nm)	(nm)		(%)	(m^2^/g)	(mmol/g)	Content *	Content *	Content *	Content *	Content *
SWCNTs	NM-411		2 ^a^	Pristine ^a^	93.1	861	0.89					
	NRCWE-055		1–2	Pristine	91.9	453.1	1.55	4.39	1.33	0.04	0.029	0.027
	NRCWE-056		1–2	-OH	89.6	356.7	2.76	1.26	3.65	0.1	0.04	0.026
	NRCWE-057		1–2	-COOH	83.1	281.6	6.02	2.2	2.74	0.14	0.177	0.044
	NRCWE-051		1–2	Pristine	90.2	442.6	1.13	1.63	1.08	0.06	0.041	-
	NRCWE-052		1–2	Pristine	92.9	405.7	0.83	1.05	1.23	0.12	0.028	0.009
	NRCWE-053		1–2	-OH	88.2	367.8	1.85	0.85	3.82	0.1	0.021	0.023
	NRCWE-054		1–2	-COOH	87.9	370.8	3.03	1.59	3.81	0.13	0.027	-
MWCNTs	NRCWE-061	730.85	16.42	-NH_2_	96.5	170.4	0.42	0.59	0.001	1.97	0.01	0.005
	NRCWE-062	468	8.82	Pristine	89.1	443.2	2.59	0.58	4.6	0.22	0.074	-
	NRCWE-063	345.35	14.18	-OH	88.2	426.4	2.66	1.95	5.87	0.49	0.072	0.023
	NRCWE-064	213.6	7.46	-COOH	88.4	445.2	3.21	1.8	5.51	0.3	0.027	0.028

The fiber lengths were determined via the computerized image analysis of SEM micrographs. OH is the amount of functionality assuming that all oxygen measured with combustion elemental analysis belonged to OH groups. NM-411 is available in the European Commission’s Joint Research Center nanomaterial repository. ^a^ Characteristics are reported in the public EUFP7 NANoREG database: https://search.data.enanomapper.net/ (accessed on 18 October 2022) [16] * Determined with XRF. Chemical composition data were calculated as wt% of all the oxides measured. -: Not detected. The lengths and diameter of the SWCNTs could not be measured because of their tangled morphology.

## Data Availability

The data presented in this study are available on request from the corresponding author.

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
