# Peer review of "Single-Walled vs. Multi-Walled Carbon Nanotubes: Influence of Physico-Chemical Properties on Toxicogenomics Responses in Mouse Lungs"

_nanomaterials, 2023, doi:10.3390/nano13061059_

Round 1
Reviewer 1 Report
Abstract: The results should be more clearly presented
Results should be condensed and better presented.
How big is the experimental error?
There is no clear characterization. I propose to replave table 1 by a figure.
Clear conclusions is missing
Reviewer 2 Report
Congratulation for very good manuscript preparation with very high level, good Figures, excellent characterization and well discussion
I have some small observations
- In the introduction part mentioned “were evaluated at 1 and 28 days post-exposure” why selected this period of time
- 2. Materials and Methods, including information about purification materials or sentence “used as received”
- Figures 7 and 12, will be difficult to read the information in the final manuscript version, try to increase the words size
- Improve figures quality at minimal 300 dpi
- Try to improve discussion part
- Manuscript has only two references from 2022, include more recently references
Reviewer 3 Report
This article is about the changes in lung gene expression and changes in tissue induced by single and multi-wall carbon nanotubes. The work contains really a huge amount of data and is very valuable for the society. I suggest a minor revision prior to publication.
1 1 Reference of sentence 1 missing.
2. 2 Reference of sentence 2 missing.
3. 3 Providers of the organic coating chemicals and other processing chemicals should be mentioned. Not only the carbon nanotubes.
4. 4 How were the CNT’s modified? If they are modified purchased this should be clearly stated out.
5. 5 Line 284 grammar: There are more than 12 CNTs in this test.
6. 6 Figure 6 a and b have different letter size
7. 7 Figure 7: letters are really small and hard to read.
8. 8 Figure S9: Letters are sooo small, I can not read them, please choose a presentation that can allow the readers to read the information, having longer SI is fine for this important information.
9. 9 Figure 12 has really too small X axis labels. Please increase or re-structure figure.
110. The last supporting figure I see in supporting .pdf is Figure S10 and not S11, please add it.
111. The authors report problems of isolation of the nanotubes. The group of Nakashima in Japan solved this problem decades ago.[1]
112. There are composite approaches for nanotubes as sensors for mechanically stressed polylectrolyte systems[2]. Polyelectrolytes are also simplified models for proteins,[3] do the authors think, this could be like a model for influencing the signalling pathway? Also the lung is a mechanically stressed system which might help uncover the cause of the influenced signalling pathways.
References
[1] Y. Tomonari, H. Murakami, and N. Nakashima, “Solubilization of single-walled carbon nanotubes by using polycyclic aromatic ammonium amphiphiles in water--strategy for the design of high-performance solubilizers.,” Chem. Eur. J., vol. 12, no. 15, pp. 4027–34, May 2006, doi: 10.1002/chem.200501176.
[2] J. Frueh, N. Nakashima, Q. He, and H. Möhwald, “Effect of Linear Elongation on Carbon Nanotube and Polyelectrolyte Structures in PDMS-Supported Nanocomposite LbL Films,” J. Phys. Chem. B, vol. 116, no. 40, pp. 12257–12262, Oct. 2012, doi: 10.1021/jp3071458.
[3] P. M. Visakh and O. Bayraktar, Polyelectrolytes Thermodynamics and Rheology, 1st ed. Heidelberg: Springer International Publishing, 2014. doi: 10.1007/978-3-319-01680-1.
